# Exploring perception and attitude of nursing students towards interprofessional education in Saudi Arabia

Zeinab A. Abusabeib[1]*, Nadiah A. Baghdadi[2], Noura A. Almadni[1], Hala K. Ibrahim[1]

1 Department of Community Health Nursing and Psychiatric and Mental Health, College of Nursing, Princess Nourah bint Abdulrahman University, Riyadh, Saudi Arabia, 2 Department of Nursing Management and Education, College of Nursing, Princess Nourah Bint Abdulrahman University, Riyadh, Saudi Arabia

☯ These authors contributed equally to this work.
* zaabusabeib@pnu.edu.sa

## Abstract

### Introduction

Interprofessional education is a process designed to facilitate knowledge exchange between healthcare professionals with the aim of improving learning, collaboration, and patient care. It helsps students gain a better understanding of communication, teamwork, and each profession's role. This study aims to examine Saudi undergraduate nursing students' attitudes and readiness for engagement in high-fidelity simulation interprofessional education and practice after training nurse educators in HFS.

### Methodology

A descriptive cross-sectional study was conducted among 311 nursing students. The process of sampling was done by convenience and was not probabilistic. Participants were asked to complete a self-administered questionnaire. The survey included demographic information as well as the Readiness for Interprofessional Learning Scale.

### Results

There was a positive attitude toward Interprofessional education among nursing students and a reasonable level of readiness. The Readiness for Interprofessional Learning Scale score was significantly associated with academic years ($p \leq 0.05$).

### Conclusion

Nursing students have a positive attitude towards interprofessional education and are willing to engage in high-fidelity simulation activities. The integration of high-fidelity simulation in interprofessional education can significantly benefit nursing students by enhancing their clinical skills, decision-making abilities, and teamwork dynamics in a controlled environment.

**Data Availability Statement:** All relevant data are within the manuscript and its Supporting Information files. This submission contains all raw data required to replicate the results of your study.

**Funding:** This work is going to be supported by Princess Noura bint Abdul Rahman University Deanship of Scientific Research. The funders had no role in study design, data collection and analysis, decision to publish, or preparation of the manuscript.

**Competing interests:** NO authors have competing interests

# Introduction

Healthcare professional education is moving towards the implementation of Interprofessional education (IPE) for both undergraduate and postgraduate studies on an international level [1]. IPE is an educational process that facilitates the exchange of knowledge between healthcare professionals for the purpose of improving learning and collaboration [2]. There has been a substantial amount of attention given to IPE in the World Health Organization [3]. In many health-related professions education programs, IPE has become part of the accreditation requirements [4]. A collaborative effort between healthcare professionals has been effective in reducing medical errors, improving patient satisfaction, enhancing patient care, and improving knowledge and skills. There are typically a number of health professionals involved in the interdisciplinary healthcare team, including physicians, nurses, pharmacists, therapists, and dietitians [5].

IPE addresses the challenges of inter-professional conflict and misunderstanding by fostering respect and understanding among healthcare teams. Due to the sharing of knowledge between students, students are better able to understand each other's capabilities, and their team identity is strengthened as a result [6].

In order to ensure effective clinical practice, IPE is essential, but it remains a challenge to implement. It may be possible to overcome the time and space challenges associated with in-person interprofessional simulation by using virtual simulations as part of IPE activities [7]. Among the pedagogies of IPE, high-fidelity simulation-based learning (SBL) can be an effective method of implementing IPE [8]. Previous research has emphasized the importance of IPE in promoting teamwork, communication, and mutual respect among healthcare professionals [7]. However, there is limited research on nursing students' attitudes and readiness towards high-fidelity simulation (HFS) IPE, specifically in the context of Saudi Arabia. HFS provides nursing students with a practical and immersive learning environment that can enhance their clinical skills, decision-making abilities, and teamwork dynamics [9]. In designing effective educational interventions and improving collaborative practice in the Saudi healthcare system, it is vital to understand how nursing students perceive and approach IPE through HFS.

This study is unique in its focus on Saudi undergraduate nursing students' attitudes toward HFS IPE by well-trained educators, contributing to the growing literature on IPE in the Saudi context. Using HFS adds a practical and experiential dimension to the study, providing insights into how nursing students perceive and interact with interprofessional teams in simulated clinical settings. The healthcare students must be willing to learn together in order for the IPE to have a positive effect on inter-professional collaboration [10, 11]. Attitude factors have been identified as a major barrier to the effective implementation of it. Some of the challenges and obstacles facing IPE include lack of support, and resources including faculty members and financial resources [12, 13].

Moreover, large student numbers constitute a barrier to carrying out learning activities at the same time. Timetabling difficulties, differences in the length of professional programs, various assessment methods employed by different disciplines, and difficulties relating to planning and resource allocation are also some of the obstacles identified [14]. Overcoming these obstacles is by adequately researching and exploring the different ways in which IPE can be introduced and developed, strengthening interprofessional practice, and working on joint decision-making [15].

Understanding nursing students' attitudes and readiness toward IPE is essential in Saudi Arabia, where the healthcare system is rapidly evolving. As Saudi Arabia invests more in healthcare education and workforce development, understanding nursing students' attitudes

towards IPE and HFS can help inform curriculum design, faculty development, and student support initiatives. Also, by training nurse educators on HFS, they can design scenarios that are both realistic and aligned with the nursing curriculum's learning objectives. This approach is crucial as it ensures that educators are proficient in using simulation technology and techniques effectively in IPE, ultimately enhancing the learning experience for students [16]. Addressing this gap provides a valuable perspective for enriching nursing IPE through simulation, and ultimately improve patient care outcomes in Saudi Arabia.

This study aims to examine Saudi undergraduate nursing students' attitudes and readiness for engagement in high-fidelity simulation interprofessional education and practice after training nurse educators in HFS.

## Methods

### Study design, setting, and population

A program was created to train nurse educators on how to effectively instruct nursing students in IPE sessions. This program consisted of a series of workshops. An assessment was conducted to evaluate the effectiveness of this training program on nurse educators' ability to effectively teach nursing students.

During the 2023–2024 academic year, a descriptive cross-sectional research design was carried out on all third- and fourth-year undergraduate nursing students (320) at Princess Noura bint Abdul Rahman University (PNU). The process of sampling was done by convenience and was not probabilistic. First- and second-year students were excluded from the study since they do not have courses in the simulation laboratory and are not aware of the full scope of their responsibilities. Additionally, those who did not volunteer to participate were also excluded.

The 320 nursing students' sample size was based on the entire population of third- and fourth-year undergraduate nursing students at PNU. Including all nursing students taking courses in the simulation laboratory ensures a strong representation of the target population and have enough statistical power to identify significant differences and trends in attitudes and perceptions. This will enhance the validity of the findings. About 311 students out of 320 agreed to participate in the research. The faculty administration provided written consent for the gathering of student data.

### Research instrument

A validated two-part self-administered online questionnaire was distributed via google form at the end of the academic year 2023 following a pilot study. The first part is demographic data, including age and academic level of education. The second part includes the Readiness for Interprofessional Learning Scale (RIPLS) [17]. RIPLS contains 19 items intended to assess whether students are capable of interacting with students from other health colleges and sharing their knowledge with them. The first domain of the RIPLS is about teamwork and collaboration (items 1–9), professional identity (items 10–16), and then roles and responsibilities (items 17–19). A five-point Likert scale was used to assess students' attitudes. A score of 1 indicates strongly disagree, and a score of 5 indicates strongly agree. A mean score was calculated for each item on the scale, and the total mean score was then calculated by adding all the item scores together.

### Data analysis

In order to analyze the data, Statistical Package for Social Sciences (SPSS) version 20.0 was used. Frequency and percentages were used to describe demographic characteristics. The

Kolmogorov-Smirnov test was used to verify distribution normality. Quantitative data were described based on a range (minimum and maximum), mean, and standard deviation. Statistical significance was determined by $p < 0.05$.

To assess the clarity, length, and completion time of the questionnaire, 20 students who were not included in the study sample were given the questionnaire. The questionnaire was evaluated for its content validity, and the internal consistency of its items was determined by Cronbach's Alpha. Overall, the questionnaire had a reliability coefficient of 0.867 (Table 1).

To compare attitudes and readiness towards IPE and practice at two education levels, the overall mean scores were analyzed. Student t-test was used for normally distributed quantitative variables to determine the difference between attitudes and perceptions of students based on demographic variables. For some items, the code for responses was reversed so that higher scores reflected higher levels of Readiness for Interprofessional Learning.

## Ethical consideration

The PNU/Institutional Research Board (IRB) ethical approval was obtained. Additionally, informed consent was obtained from the participants via the self-administered online questionnaire and documented, and the participants were assured that their data would only be used for research purposes, and confidentiality was maintained throughout the study. There was no restriction on the subject's right to withdraw from the study. In addition, this research study adhered to the principles of the Declaration of Helsinki 1995.

## Results

The response rate in this study was 97.2%. The majority of participants (n = 266, 85.5%) were between the ages of 19 and 21, while 14.5% were over 21. In terms of academic level, nearly equal percentages of students are in their third (n = 157, 50.5%) and fourth (n = 154, 49.5%) years of study. Table 2 presents the summary of the overall perception of knowledge on RIPLS among nursing students.

### Teamwork and collaboration

Close inspection of Table 2 shows that most of the participants (77.5%) either "strongly agreed" or "agreed" that to solve patient problems, healthcare students should work together. More than 82% expressed that shared learning with other healthcare students would increase their ability to understand clinical problems. About 85.9% indicated and believed that for small groups to learn to work, students need to trust and respect each other. Many participants (79.7%) "strongly agreed" or "agreed" that teamwork skills are essential for all healthcare students to learn.

### Professional identity

It is apparent from Table 2 that about one-quarter of the participants agreed that they do not want to waste their time learning alongside other healthcare students, while 56.9% of the

**Table 1. Reliability statistics.**

| RIPS Scale domains | Cronbach's Alpha | No. of Items |
|---|---|---|
| Teamwork & Collaboration | 0.941 | 9 |
| Professional Identity | 0.720 | 7 |
| Roles And Responsibility | 0.621 | 3 |
| Overall Readiness for Interprofessional Learning Scale (RIPLS) | 0.867 | 19 |

Table 2. Distribution of nursing students according to the overall perception of knowledge on Readiness for Interprofessional Learning Scale (RIPLS) items.

| Q | Overall Readiness for Interprofessional Learning Scale (RIPLS) | Strongly disagree | Disagree | Neutral | Agree | Strongly agree | Mean ± SD. |
|---|---|---|---|---|---|---|---|
| | | No. (%) | No. (%) | No. (%) | No. (%) | No. (%) | |
| 1 | **Teamwork & Collaboration** | | | | | | |
| 1 | **Learning with other students will help me become a more effective member of a healthcare team** | 17 (5.5%) | 10 (3.2%) | 51 (16.4%) | 93 (29.9%) | 140(45.0%) | **4.06 ± 1.11** |
| 2 | **Patients would ultimately benefit if health-care students worked together to solve patient problems** | 18 (5.8%) | 8 (2.6%) | 44 (14.1%) | 108 (34.7%) | 133 (42.8%) | **4.06 ± 1.09** |
| 3 | **Shared learning with other health-care students will increase my ability to understand clinical problems** | 16 (5.1%) | 13 (4.2%) | 25 (8.0%) | 108 (34.7%) | 149 (47.9%) | **4.16 ± 1.08** |
| 4 | **Learning with health-care students before qualification would improve relationships after qualification** | 12 (3.9%) | 15 (4.8%) | 57 (18.3%) | 124 (39.9%) | 103 (33.1%) | **3.94 ± 1.03** |
| 5 | **Communication skills should be learned with other health-care students** | 12 (3.9%) | 16 (5.1%) | 52 (16.7%) | 116 (37.3%) | 115 (37.0%) | **3.98 ± 1.05** |
| 6 | **Shared learning will help me to think positively about other professionals** | 15 (4.8%) | 14 (4.5%) | 48 (15.4%) | 116 (37.3%) | 118 (37.9%) | **3.99 ± 1.07** |
| 7 | **For small group learning to work, students need to trust and respect each other** | 14 (4.5%) | 8 (2.6%) | 22 (7.1%) | 78 (25.1%) | 189 (60.8%) | **4.35 ± 1.03** |
| 8 | **Team-working skills are essential for all health care students to learn** | 16 (5.1%) | 14 (4.5%) | 33 (10.6%) | 93 (29.9%) | 155 (49.8%) | **4.15 ± 1.11** |
| 9 | **Shared learning will help me to understand my own limitations** | 14 (4.5%) | 13 (4.2%) | 57 (18.3%) | 107 (34.4%) | 120 (38.6%) | **3.98 ± 1.07** |
| 2 | **Professional Identity** | | | | | | |
| 10 | **I don't want to waste my time learning with other health care students Strongly** | 80 (25.7%) | 97 (31.2%) | 59 (19.0%) | 38 (12.2%) | 37 (11.9%) | **3.47 ± 1.31** |
| 11 | **It is not necessary for undergraduate health-care students to learn together** | 89 (28.6%) | 82 (26.4%) | 67 (21.5%) | 39 (12.5%) | 34 (10.9%) | **3.49 ± 1.32** |
| 12 | **Clinical problem-solving skills can only be learned with students from my own department** | 68 (21.9%) | 84 (27.0%) | 74 (23.8%) | 46 (14.8%) | 39 (12.5%) | **3.31 ± 1.31** |
| 13 | **Shared learning with other health-care students will help me to communicate better with patients and other professionals** | 14 (4.5%) | 12 (3.9%) | 49 (15.8%) | 103 (33.1%) | 133 (42.8%) | **4.06 ± 1.07** |
| 14 | **I would welcome the opportunity to work on small-group projects with other health-care students** | 16 (5.1%) | 19 (6.1%) | 62 (19.9%) | 114 (36.7%) | 100 (32.2%) | **3.85 ± 1.10** |
| 15 | **Shared learning will help to clarify the nature of patient problems** | 17 (5.5%) | 9 (2.9%) | 47 (15.1%) | 132 (42.4%) | 106 (34.1%) | **3.97 ± 1.05** |
| 16 | **Shared learning before qualification will help me become a better team worker** | 16 (5.1%) | 11 (3.5%) | 55 (17.7%) | 108 (34.7%) | 121 (38.9%) | **3.99 ± 1.08** |
| 3 | **Roles And Responsibility** | | | | | | |
| 17 | **The function of nurses and therapists is mainly to provide support for doctors** | 62 (19.9%) | 53 (17.0%) | 67 (21.5%) | 70 (22.5%) | 59 (19.0%) | **2.96 ± 1.40** |
| 18 | **I'm not sure what my professional role will be** | 63 (20.3%) | 91 (29.3%) | 71 (22.8%) | 52 (16.7%) | 34 (10.9%) | **3.31 ± 1.27** |
| 19 | **I have to acquire much more knowledge and skills than other health-care students** | 16 (5.1%) | 28 (9.0%) | 88 (28.3%) | 90 (28.9%) | 89 (28.6%) | **2.33 ± 1.13** |

participants disagreed. About 75.9% agreed that shared learning with other healthcare students will help them to communicate better with patients and other professionals. More than 73% agreed that shared learning before qualification will help them become a better team worker.

## Roles and responsibility

The results, as shown in Table 2, indicate that 41.5% agreed that the function of nurses and therapists is mainly to provide support for doctors. About half of the participants indicated

**Table 3. Descriptive analysis of the nursing students according to scores for overall Readiness for Interprofessional Learning Scale (RIPLS) (n = 311).**

| Overall Readiness for Interprofessional Learning Scale (RIPLS) | Total score | | | Average score (1–5) |
|---|---|---|---|---|
| | Score Range | Min.–Max. | Mean ± SD. | Mean ± SD. |
| Teamwork & Collaboration | (9–45) | 9.0–45.0 | 36.67 ± 7.91 | 4.07 ± 0.88 |
| Professional Identity | (7–35) | 11.0–35.0 | 26.13 ± 4.85 | 3.73 ± 0.69 |
| Roles And Responsibility | (3–15) | 3.0–15.0 | 8.61 ± 2.92 | 2.87 ± 0.97 |
| Overall | (19–95) | 33.0–95.0 | 71.41 ±11.38 | 3.76 ± 0.60 |

they are sure what their professional role will be. About 58% agreed they must acquire much more knowledge and skills than other healthcare students.

Table 3 provides summary statistics for the descriptive analysis of nursing students based on their overall RIPLS scores. The overall mean score for RIPLS was (71.41 ±11.38) with an average of (3.76 ± 0.60).

Table 4 illustrates the relationship between the total score for the overall RIPLS and demographic data (n = 311). Academic years were significantly associated with the mean total score for RIPLS (p ≤ 0.05). The total score for RIPLS increases as the number of academic years increases.

## Discussion

With this era of rapid development, if the foundations of IPE and practice are laid during the university years of students in simulation labs, the healthcare system will gain the visibility of well-trained healthcare professionals [9].

There is a strong interest among nursing students to explore interprofessional collaboration and learning opportunities, as evidenced by the high participation rate of 311 students in the study. Based on the findings of this study, students' scores were high across 19 RIPLS items, with the exception of a small minority of questions. This indicates that IPE was perceived positively by participants from the College of Nursing at Princess Nourah bint Abdulrahman University and was associated with both benefits and challenges. This result also shows that the students are well-prepared for the implementation of IPE activities and shared learning. The findings of this study are consistent with the positive results reported from previous studies in the related literature [18–20].

In this study, the items in subscale one related to teamwork and collaboration were ranked highest by all respondents. This shows that respondents strongly believe that shared learning is

**Table 4. The association between demographics' characteristics and RIPLS.**

| Demographic data | Total score for RIPLS | t | P |
|---|---|---|---|
| | Mean ± SD. | | |
| Age | | | |
| 19–21 | 71.45 ± 11.24 | 0.187 | 0.852 |
| > 21 | 71.11 ± 12.30 | | |
| Academic year | | | |
| 3rd | 69.30 ± 11.56 | 3.350* | 0.001* |
| 4th | 73.55 ± 10.81 | | |

SD: Standard deviation t: Student t-test

p: p value for comparison between the studied categories

*: Statistically significant at p ≤ 0.05.

beneficial in several ways. It was found that a large percentage of students (74%) considered learning with other students to be an effective method of becoming a member of a healthcare team. As reported in a previous study conducted among nursing students in Lebanon, the findings are in agreement with the present study [21].

A further finding of the study indicated that 82.6% of the students believed that sharing learning with other healthcare students would enhance the clinical problems understanding. In this way, knowledge and skills can be shared with other undergraduates as a means of gaining a deeper understanding of clinical problems that occur in the workplace. Compared to studies conducted in Germany to assess IPE perception among students, these results are consistent [22]. Also, a study conducted in China demonstrated that the incorporation of interprofessional education and simulation in the curriculum had a beneficial effect on the knowledge and skills of undergraduate nursing students [23].

Collaborative learning is beneficial for students and results in gaining better communication skills as well as a more positive attitude toward other professionals. According to the results of the study, 74.3% of the students believed that communication skills should be acquired with other healthcare students. Ho JM et al. found similar results among nursing and physiotherapy students [24]. An additional study conducted among medical and nursing students in the United States has confirmed the findings of this study [25].

Study findings from Norway and the United States, however, suggest that communication with other professions has not improved significantly, and students are reluctant to express concerns [26, 27]. This finding of the current study may be inconsistent with these mentioned studies because IPE outcomes are dependent on the learning environment design. As part of IPE, clinical exchange and team collaboration must be balanced. Scenarios should be designed by educators that enables all professions to perform their duties equally [28].

In addition to addressing emerging healthcare issues, solving problems, and delivering services to populations, teamwork will help students better understand each profession's role in patient care and health, as well as add value and significance to each other. Teamwork skills are essential for all healthcare students, according to 79.7% of respondents. A study conducted among healthcare students revealed that participants understand that teamwork plays a significant role in positive outcomes [29].

The items in second part of the questionnaire relate to both positive and negative aspects of professional identity. According to the majority of students participating in this study, shared SBL with other healthcare students would facilitate better communication with patients and other professionals and clarify the nature of patients' problems. Also, it was agreed that shared learning before qualification would increase their ability to work as a team. According to studies conducted in the United States and Iran, shared learning positively impacts students' attitudes and enhances their professional identities [30, 31].

According to the subscale on Roles and responsibilities, there is a 41.5% consensus in this study that nurses and other therapists are primarily responsible for supporting physicians. Recently, however, there has been a shift in emphasis toward the concept that leadership should be determined by the context in which the team operates. A total of 27.6% of respondents indicated that they are not certain of their future professional role. It is expected that the IPE experience will enable them to gain a greater understanding of both the importance of the other professions' roles as well as the importance of their own roles in the care of patients. Several studies have indicated that students had a greater understanding of the role of healthcare professionals after participating in an IPE experience [32–34].

It was found that 57.5% of the students agreed they would need to acquire much more knowledge and skills than other healthcare students. However, in a study conducted in New

Zealand, medical students agreed that a greater level of expertise and knowledge is required of them than nursing students or pharmacy students [35].

According to a comparison of the mean scores of the three subscales, it was evident that students acknowledged the value of IPE, as evidenced by the subscale scores on teamwork and collaboration, as well as professional identity, however, students scored comparatively lower on the role and responsibility subscale.

The study findings indicated no statistically significant difference according to the age of nursing students. However, the academic year was significantly related to the mean total score for RIPLS (p ≤ 0.05). The higher the educational level, the greater the RIPLS score. This finding goes against the outcome of a study that found that students in various undergraduate healthcare specializations scored lower on RIPLS subscales as they neared the final year of education [36]. It also contradicts a study that found no significant difference in students' perceptions of IPE based on their year of study [37]. In this study, the shift in students' attitudes towards IPE becoming more positive as they advanced in their education indicated their willingness to learn and work collaboratively. For educators, this suggests that introducing IPE curricula early in undergraduate education could strengthen students' positive attitudes towards such a system.

## Study limitations

The study's high response rate was seen as a significant strength, but its limitation lies in the fact that it only involved nursing students from a one university, which limits the generalizability of the findings. Nevertheless, each participant offered a distinct viewpoint on IPE. In addition, in this study, selection was based on convenience and not probability, which could introduce bias and affect the representativeness of the sample.

The research also uses a descriptive and cross-sectional approach, which precludes the tracking of changes over time.

## Implications for future research and practice

More information regarding the sustainability of interprofessional education on nursing students' attitudes can be obtained through longitudinal studies. Also, among the recommendation for future researchers is that students from other health professions at PNU should be involved in the IPE process at the simulation center/King Abdullah bin Abdulaziz University Hospital (KAAUH), as involving students from other health disciplines will enhance the quality of the IPE experience.

The study results can be used to develop interprofessional education programs in Saudi Arabia and tailor interventions that address areas of concern. Furthermore, faculty members should be provided with effective training to facilitate such activities. The study can help identify areas where faculty members may need additional support. The findings can be used to develop strategies that promote collaboration, communication, and teamwork among nursing students and other healthcare professionals.

From an economic perspective, the implementation of IPE has the potential to result in a more effective utilization of resources in healthcare education. Through encouraging cooperation among students from various health professions, educational institutions can make the most of training facilities, decrease duplication in training programs, and ultimately reduce costs related to separate training sessions. Additionally, successful interprofessional collaboration has been associated with better patient outcomes, which can lead to decreased healthcare expenses in the long term due to a decrease in errors, shorter hospital stays, and increased patient satisfaction.

## Conclusions

Nursing students in Saudi Arabia were found to have a positive attitude towards interprofessional education and a willingness to engage in high-fidelity simulation activities based on the study's results.

Overcoming the challenges of implementing interprofessional education and integrating high-fidelity simulation can ensure sustainability and be extremely beneficial for nursing students. It facilitates an immersive and realistic learning experience that narrows the gap between theoretical knowledge and clinical practice. Through participation in high-fidelity simulation scenarios, students can enhance their clinical skills, decision-making abilities, and teamwork dynamics in a secure and controlled environment.

The outcomes of this research indicate that incorporating high-fidelity simulation into interprofessional education courses can help tackle the challenges posed by limited clinical placement opportunities and varying levels of clinical exposure among nursing students. By offering students opportunities to practice their skills and work collaboratively with other healthcare professionals in simulated exercises, well- trained instructors can better equip them to navigate the intricacies of contemporary healthcare delivery. Addressing this perspective enriches nursing IPE through simulation and improves patient care outcomes in Saudi Arabia.

## Supporting information

**S1 Raw data.**
(PDF)

**S2 Raw data.**
(XLSX)

## Acknowledgments

Researchers would like to thank and appreciate all participants who voluntarily completed the survey and shared their experiences.

## Author Contributions

**Conceptualization:** Zeinab A. Abusabeib, Nadiah A. Baghdadi, Noura A. Almadni, Hala K. Ibrahim.

**Data curation:** Zeinab A. Abusabeib, Nadiah A. Baghdadi, Noura A. Almadni, Hala K. Ibrahim.

**Formal analysis:** Zeinab A. Abusabeib, Nadiah A. Baghdadi, Noura A. Almadni, Hala K. Ibrahim.

**Funding acquisition:** Zeinab A. Abusabeib, Nadiah A. Baghdadi, Noura A. Almadni, Hala K. Ibrahim.

**Investigation:** Zeinab A. Abusabeib, Nadiah A. Baghdadi, Noura A. Almadni, Hala K. Ibrahim.

**Methodology:** Zeinab A. Abusabeib, Nadiah A. Baghdadi, Noura A. Almadni, Hala K. Ibrahim.

**Project administration:** Zeinab A. Abusabeib, Nadiah A. Baghdadi, Noura A. Almadni, Hala K. Ibrahim.

**Resources:** Zeinab A. Abusabeib, Nadiah A. Baghdadi, Noura A. Almadni, Hala K. Ibrahim.

**Software:** Zeinab A. Abusabeib, Nadiah A. Baghdadi, Noura A. Almadni, Hala K. Ibrahim.

**Supervision:** Zeinab A. Abusabeib, Nadiah A. Baghdadi, Noura A. Almadni, Hala K. Ibrahim.

**Validation:** Zeinab A. Abusabeib, Nadiah A. Baghdadi, Noura A. Almadni, Hala K. Ibrahim.

**Visualization:** Zeinab A. Abusabeib, Nadiah A. Baghdadi, Noura A. Almadni, Hala K. Ibrahim.

**Writing – original draft:** Zeinab A. Abusabeib, Nadiah A. Baghdadi, Noura A. Almadni, Hala K. Ibrahim.

**Writing – review & editing:** Zeinab A. Abusabeib, Nadiah A. Baghdadi, Noura A. Almadni, Hala K. Ibrahim.

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
