## [Decision Letter · Decision Letter 0]

29 Apr 2024

PONE-D-24-01945Exploring Perception and Attitude of Nursing Students Towards Interprofessional Education in Saudi ArabiaPLOS ONE

Dear Dr. Abu Sabeib,

Thank you for submitting your manuscript to PLOS ONE. After careful consideration, we feel that it has merit but does not fully meet PLOS ONE’s publication criteria as it currently stands. Therefore, we invite you to submit a revised version of the manuscript that addresses the points raised during the review process.

We look forward to receiving your revised manuscript.

Kind regards,

Naeem Mubarak, PhD

Academic Editor

PLOS ONE

Journal Requirements:

https://www.frontiersin.org/articles/10.3389/fpubh.2022.1030863/full

In your revision ensure you cite all your sources (including your own works), and quote or rephrase any duplicated text outside the methods section. Further consideration is dependent on these concerns being addressed.

We will receive funding from Princess Nourah bin Abdulrahman University, Riyadh, Saudi Arabia

5. We note that your Data Availability Statement is currently as follows: All relevant data are within the manuscript and its Supporting Information files

Additional Editor Comments:

The manuscript requires major revisions to improve its methodology and results section in particular. To draw comparisons of similar studies focusing on interprofessional education and signify the need to integrate module of collaborative care into the curriculum; you may cite the following studies in discussion section (This is optional and should only be taken as a suggestion for the improvement of the manuscript):

DOI: 10.1136/bmjopen-2023-079507

DOI: 10.3390/antibiotics10101204

To further strengthen the evidence of how collaborative care is achieving success in other professions linked with patient care ; you may cite the following studies in discussion section (This is optional and should only be taken as a suggestion for the improvement of the manuscript):

doi: 10.3389/fpubh.2024.1323102

Reviewers' comments:

Reviewer's Responses to Questions

**Comments to the Author**

1. Is the manuscript technically sound, and do the data support the conclusions?

Reviewer #1: Yes

Reviewer #2: No

2. Has the statistical analysis been performed appropriately and rigorously? 

Reviewer #1: I Don't Know

Reviewer #2: No

3. Have the authors made all data underlying the findings in their manuscript fully available?

Reviewer #1: No

Reviewer #2: Yes

4. Is the manuscript presented in an intelligible fashion and written in standard English?

Reviewer #1: Yes

Reviewer #2: Yes

5. Review Comments to the Author

**Reviewer #1:** I would like to, firstly, congratulate the authors for their effort and dedication in writing this scientific article. I hope that the following considerations will further strengthen your work.

In the introduction, it would be pertinent to include the current state-of-the-art regarding nursing students' readiness and attitudes for IPE. We have existing studies that have analyzed the attitude and readiness of nursing and healthcare students in simulation scenarios. Therefore, I suggest addressing existing studies in the field, the uniqueness of your study, and/or the relevance of your study, considering the simulation scenario. Students underwent a high-fidelity simulation cenario, and I noticed that it was not adequately explored throughout your scientific article, even though it was integrated into your aim. Integrating this aspect into both the discussion and conclusion sections would be beneficial, as it is through this pedagogical intervention that readiness and attitudes for IPE was analyzed, thus serving as a important component in the interpretation of the gathered data.

In the methodology, in the "Study design, Setting, and population" section, it would be useful to characterize/describe the university and, if possible, the nursing course. It is important in the methodology to also describe the high-fidelity simulation, as it is an important element, being the pedagogical intervention used to assess attitudes and readiness for interprofessional education and practice. In line 88, I suggest describing the name of the platform used to distribute online questionnaires to participants.

In the methodology, between lines 93 and 94, I wondered if validation occurred through the pilot study described later? If so, it's pertinent to indicate the pilot study here. For example: "A validated self-administered online questionnaire was administered following a pilot study." It is also pertinent to change the words to avoid repetition of "self-administered" and "administered."

In the discussion, in line 180, I noticed that two of these cited studies present results from faculty members and educators. It would be interesting to cite and relate to your results studies that examined attitudes using the RIPLS in nursing students during realistic simulation. For example: (This is optional and should only be taken as a suggestion for the improvement of the manuscript): 

Examining interprofessional learning perceptions among students in a simulation-based operating room team training experience. DOI: 10.1080/13561820.2018.1513464 

Does Interprofessional Scenario-Based Simulation Training Change Attitudes Towards Interprofessional Learning - A Pretest-Posttest Study. DOI: 10.2147/JMDH.S370100

Effects of High-Fidelity Simulation on Physical Therapy and Nursing Students' Attitudes Toward Interprofessional Learning and Collaboration. DOI: 10.3928/01484834-20170712-03

Effects of a single interprofessional simulation session on medical and nursing students' attitudes toward interprofessional learning and professional identity: a questionnaire study. DOI: 10.1186/s12909-020-1971-6

Implementation and evaluation of an interprofessional simulation-based education program for undergraduate nursing students in operating room nursing education: a randomized controlled trial. DOI: 10.1186/s12909-015-0400-8

In the discussion, between lines 200 and 203, we note that realistic simulation is an important aspect to consider in the results you identified regarding attitude and readiness in nursing students.

What is the reference/citation for the passage in the discussion "Recently, however, there has been a shift in emphasis toward the concept that leadership should be determined by the context in which the team operates"?

In line 226, it is appropriate to clarify in the writing that it refers to "nursing students."

In your discussion, in line 235, how does this finding "however, the academic year was significantly related to the mean total score for RIPLS (p ≤ 0.05)” relate to what we have in the scientific literature? In line 241, replace the comma with a period.

In your conclusions, it is important to include information about the high-fidelity simulation scenario, as there was an interesting connection in the discussion with the use of the simulation. This consideration also applies to the conclusion in your abstract.

**Reviewer #2:** Dear Authors,

Thanks for choosing PLOS ONE. I hope my feedback is helpful for you.

Abstract: please add the sampling method (i.e., convenience sampling method).

Methods: Please address the reporting guidelines (i.e., STROBE). For more information, please check the link below https://www.equator-network.org

Elaborate more on the data collection procedure

The reference for the RIPLS was not addressed within the methods section. Please cite the original study and report the reliability and validity of the instrument.

Please replace this statement: " Statistical significance was determined by p ≤ .05." to " statistical significance was determined by p <.05.

I recommend you to modify the heading of Table 4 to "The association between demographics' characteristics and RIPLS"

The limitations of the study were too brief. Please address the major limitations associated with the methodology (i.e., sampling method, research design, etc...) and how these limitations affect the generalizability of the findings.

Address the implications for future research and practice in a separate heading.

6. PLOS authors have the option to publish the peer review history of their article (what does this mean?). If published, this will include your full peer review and any attached files.

Reviewer #1: No

Reviewer #2: **Yes: **Dr. Adnan Innab

---

## [Author Response · Author response to Decision Letter 0]

22 May 2024

Reviewer 1: 

I have incorporated all your suggestions into my revision. They were very helpful. Thank you.

• The statistical analysis was revised.

• All data underlying the findings fully available.

• In the introduction, few paragraphs were replaced, and paragraphs related to simulation-based interprofessional education were added. Some of the references were updated.

• In the methodology, the data collection procedure was amended, more details regarding design, setting, population, instrument, and analysis were added. Also, High-fidelity simulation methodology was added, and the consent type was added. 

• In the discussion, some of the suggested references were cited and included.

• Paragraphs related to simulation-based interprofessional education were added. Some of the suggested references were cited and included.

• Simulation was considered between mentioned lines.

• Reference for the referred passage was added.

• Clarifying in the writing that it refers to "nursing students was done.

• How findings are related to what we have in the literature was addressed. 

• The conclusion was rewritten taking into account the use of simulation.

Reviewer 2: I have incorporated all of your suggestions into my revision. Thank you for your help.

• The manuscript has been revised taking into account that it should be technically sound, and the data support the conclusion.

• The statistical analysis was revised.

• In the abstract: The sampling method was added. Conclusion was rewritten. 

• Reporting guidelines were followed. 

• The data collection procedure was amended. 

• More details regarding design, setting, population, instrument, and analysis were added.

• Methods: The reference for the RIPLS was not addressed 

‘’Statistical significance was determined by p ≤ .05." was replaced to " statistical significance was determined by p <.05.’’

Table 4 title was modified.

The consent type was added. 

• Major limitations associated with the methodology were addressed and how these limitations affect the generalizability of the findings. 

• The implications for future research and practice in a separate heading was addressed.

---

## [Decision Letter · Decision Letter 1]

1 Sep 2024

PONE-D-24-01945R1Exploring Perception and Attitude of Nursing Students Towards Interprofessional Education in Saudi ArabiaPLOS ONE

Dear Dr. Abu Sabeib,

Thank you for submitting your manuscript to PLOS ONE. After careful consideration, we feel that it has merit but does not fully meet PLOS ONE’s publication criteria as it currently stands. Therefore, we invite you to submit a revised version of the manuscript that addresses the points raised during the review process.

Kind regards,

Naeem Mubarak, PhD

Academic Editor

PLOS ONE

Journal Requirements:

Additional Editor Comments:

The manuscript may be accepted for publication after minor revisions

Reviewers' comments:

Reviewer's Responses to Questions

**Comments to the Author**

1. If the authors have adequately addressed your comments raised in a previous round of review and you feel that this manuscript is now acceptable for publication, you may indicate that here to bypass the “Comments to the Author” section, enter your conflict of interest statement in the “Confidential to Editor” section, and submit your "Accept" recommendation.

Reviewer #3: (No Response)

Reviewer #4: All comments have been addressed

2. Is the manuscript technically sound, and do the data support the conclusions?

Reviewer #3: Yes

Reviewer #4: Yes

3. Has the statistical analysis been performed appropriately and rigorously? 

Reviewer #3: Yes

Reviewer #4: Yes

4. Have the authors made all data underlying the findings in their manuscript fully available?

Reviewer #3: Yes

Reviewer #4: Yes

5. Is the manuscript presented in an intelligible fashion and written in standard English?

Reviewer #3: Yes

Reviewer #4: Yes

6. Review Comments to the Author

Reviewer #3: Thank you for considering me to review this manuscript. The manuscript analyses the perception and attitude of nursing students towards interprofessional education (IPE). It certainly highlights the importance of IPE in nursing education for better understanding, learning, and integration in the healthcare team. However, there are few recommendations which authors can consider to improve the manuscript.

Methods

• Study Design, Setting, and Population

The authors are suggested to justify how the sample size of 320 nursing students is sufficient for the research question in line 111. Also give information about the sample size calculation in the text.

Results

The authors framed the results well. However, following points in my opinion are recommended:

• Teamwork and Collaboration

In line 155 the authors are advised to justify how they classified the 5-point Likert scale on 2-point scale of agree and disagree values. It is suggested to discuss it in text also.

Discussion

• Study Limitations

The authors are recommended to talk about the risk of bias by using convenience sampling in the text.

• Implications for Future Research and Practice

The authors discussed almost every possible implication. However, in line 276 along with IPE, the authors are recommended to talk about the economical aspect of implementation of IPE in educating the nursing students. For better insights about the interprofessional education and collaboration, the authors may refer to the following study (This is optional and should only be taken as a suggestion for the improvement of the manuscript).

1. DOI: 10.2147/RMHP.S296113

Reviewer #4: (No Response)

7. PLOS authors have the option to publish the peer review history of their article (what does this mean?). If published, this will include your full peer review and any attached files.

Reviewer #3: No

Reviewer #4: No

---

## [Author Response · Author response to Decision Letter 1]

2 Sep 2024

Reviewers' comments:

Reviewer's Responses to Questions

Comments to the Author

1. If the authors have adequately addressed your comments raised in a previous round of review and you feel that this manuscript is now acceptable for publication, you may indicate that here to bypass the “Comments to the Author” section, enter your conflict of interest statement in the “Confidential to Editor” section, and submit your "Accept" recommendation.

Reviewer #3: (No Response)

Reviewer #4: All comments have been addressed 

2. Is the manuscript technically sound, and do the data support the conclusions?

Reviewer #3: Yes

Reviewer #4: Yes 

3. Has the statistical analysis been performed appropriately and rigorously?

Reviewer #3: Yes

Reviewer #4: Yes 

4. Have the authors made all data underlying the findings in their manuscript fully available?

Reviewer #3: Yes

Reviewer #4: Yes 

5. Is the manuscript presented in an intelligible fashion and written in standard English?

Reviewer #3: Yes

Reviewer #4: Yes 

6. Review Comments to the Author

Reviewer #3: Thank you for considering me to review this manuscript. The manuscript analyses the perception and attitude of nursing students towards interprofessional education (IPE). It certainly highlights the importance of IPE in nursing education for better understanding, learning, and integration in the healthcare team. However, there are few recommendations which authors can consider to improve the manuscript.

Methods

• Study Design, Setting, and Population

The authors are suggested to justify how the sample size of 320 nursing students is sufficient for the research question in line 111. Also give information about the sample size calculation in the text .

Thank you for your suggestions. The suggested justification and information about sample size were added.

Results

The authors framed the results well. However, following points in my opinion are recommended:

• Teamwork and Collaboration

In line 155 the authors are advised to justify how they classified the 5-point Likert scale on 2-point scale of agree and disagree values. It is suggested to discuss it in text also.

Discussion 

Thank you for your suggestion. The sentences has been rewritten.

• Study Limitations

The authors are recommended to talk about the risk of bias by using convenience sampling in the text .

Thank you for your suggestion. The risk of bias has been addressed.

• Implications for Future Research and Practice

The authors discussed almost every possible implication. However, in line 276 along with IPE, the authors are recommended to talk about the economical aspect of implementation of IPE in educating the nursing students. For better insights about the interprofessional education and collaboration, the authors may refer to the following study (This is optional and should only be taken as a suggestion for the improvement of the manuscript ).

1. DOI: 10.2147/RMHP.S296113

Thank you for your suggestion. The economical aspect has been addressed.

Reviewer #4: (No Response)

7. PLOS authors have the option to publish the peer review history of their article (what does this mean?). If published, this will include your full peer review and any attached files.

Do you want your identity to be public for this peer review? For information about this choice, including consent withdrawal, please see our Privacy Policy.

Reviewer #3: No

Reviewer #4: No

---

## [Decision Letter · Decision Letter 2]

23 Sep 2024

Exploring Perception and Attitude of Nursing Students Towards Interprofessional Education in Saudi Arabia

PONE-D-24-01945R2

Dear Dr. Zeinab Abu Sabeib,

We’re pleased to inform you that your manuscript has been judged scientifically suitable for publication and will be formally accepted for publication once it meets all outstanding technical requirements.

Kind regards,

Naeem Mubarak, PhD

Academic Editor

PLOS ONE

Additional Editor Comments (optional):

The manuscript requires no further revisions

Reviewers' comments:

Reviewer's Responses to Questions

**Comments to the Author**

1. If the authors have adequately addressed your comments raised in a previous round of review and you feel that this manuscript is now acceptable for publication, you may indicate that here to bypass the “Comments to the Author” section, enter your conflict of interest statement in the “Confidential to Editor” section, and submit your "Accept" recommendation.

Reviewer #3: All comments have been addressed

2. Is the manuscript technically sound, and do the data support the conclusions?

Reviewer #3: Yes

3. Has the statistical analysis been performed appropriately and rigorously? 

Reviewer #3: Yes

4. Have the authors made all data underlying the findings in their manuscript fully available?

Reviewer #3: Yes

5. Is the manuscript presented in an intelligible fashion and written in standard English?

Reviewer #3: Yes

6. Review Comments to the Author

Reviewer #3: (No Response)

7. PLOS authors have the option to publish the peer review history of their article (what does this mean?). If published, this will include your full peer review and any attached files.

Reviewer #3: No

---

## [Editor Report · Acceptance letter]

27 Sep 2024

PONE-D-24-01945R2 

PLOS ONE

Dear Dr. Abu Sabeib, 

I'm pleased to inform you that your manuscript has been deemed suitable for publication in PLOS ONE. Congratulations! Your manuscript is now being handed over to our production team.

Kind regards, 

on behalf of

Dr Naeem Mubarak 

Academic Editor

PLOS ONE